# The Different Relationship between Homocysteine and Uric Acid Levels with Respect to the MTHFR C677T Polymorphism According to Gender in Patients with Cognitive Impairment

**DOI:** 10.3390/nu12041147

**Published:** 2020-04-19

**Authors:** Hee-Jin Kim, Il Woong Sohn, Young Seo Kim, Jae-Bum Jun

**Affiliations:** 1Department of Neurology, Hanyang University, 222, Wansimni-ro, Seondong-gu, Seoul 04763, Korea; kimys1@hanyang.ac.kr; 2Department of Rheumatology, Hanyang University Hospital for Rheumatic Diseases, 222-1, Wansimni-ro, Seondong-gu, Seoul 04763, Korea; chopin11@daum.net

**Keywords:** methylenetetrahydrofolate reductase, homocysteine, uric acid, cognitive impairment, cerebrovascular disorder

## Abstract

In an elderly population with cognitive impairment, we investigated the association between serum uric acid (sUA) and serum homocysteine (sHcy), known risk factors for cerebrovascular disease. We also investigated the potential effect of the C677T polymorphism in the gene encoding methylenetetrahydrofolate reductase (MTHFR) to the sUA level in different dementia types. Participants underwent a battery of tests including measurements of sUA, sHcy, folic acid, and vitamin B12 as well as genotyping of the MTHFR locus. Data from 861 subjects (597 females to 264 males) were retrospectively analyzed. Subjects with hyperhomocysteinemia had lower serum folic acid and vitamin B12 and higher sUA than those with normal sHcy. sUA was significantly associated with serum creatinine, HbA1c, and sHcy regardless of gender. The TT genotype was found to be associated with hyperhomocysteinemia in both genders (*p* = 0.001). The levels of hyperlipidemia, sHcy, and sUA differed according to dementia subtypes. High sUA were associated with hyperhomocystenemia in TT genotype only in dementia with vascular lesion. This study reveals that sUA is positively associated with sHcy. We speculate that the two markers synergistically increase cerebrovascular burden and suggested that dietary intervention for sUA and sHcy would be helpful for cognitive decline with vascular lesion.

## 1. Introduction

Serum uric acid (sUA) level is associated with gout, hypertension, cardiovascular disease, renal insufficiency, and metabolic syndrome [1,2]. A recent epidemiological study reported the progressive rise in the prevalence of hyperuricemia in cognitive impairment [3]. There is evidence that sUA causes endothelial dysfunction and is, therefore, associated with renal and cardiovascular disease and mortality [1,4], although others have not found such independent associations [5,6]. UA can increase radical formation and inhibit the degradation of adenosine diphosphate, producing more stable platelet aggregates and an increased incidence of thrombosis in arterial disease, especially in the patients with hypertension and diabetes [7,8,9,10,11]. Similarly, serum homocysteine (sHcy) is also involved in the disruption of vascular integrity and is well recognized as an independent risk factor for osteoporosis-associated fractures and cardiovascular and cerebrovascular diseases [12,13,14,15]. The common denominator as a surrogate marker of vascular injuries enlightened the concepts of association between sUA and sHcy [16]. Recent studies have shown a positive correlation between sUA and sHcy [17]. Both sUA and sHcy levels might be complex outcomes influenced by multiple genetic and environmental factors. The methylenetetrahydrofolate reductase (MTHFR) enzyme plays a role in both homocysteine [18] and folate metabolism [19]. Polymorphisms in the MTHFR gene have been studied in relation to various medical conditions such as cardiovascular disease, thrombosis, pregnancy complications, neural tube defects, and cancer [18,20,21]. MTHFR mutations are candidates as important genetic influences on sUA, and recent genome-wide association studies have detected genetic variants related to sUA levels [20,22]. The MTHFR genetic variant C677T was also reported as a probable independent risk factor for hyperuricemia in a meta-analysis [20,23].

No official guidelines exist regarding specific patients that should be tested for MTHFR gene polymorphisms or for homocysteine testing [20,24]. Nevertheless, hyperhomocysteinemia and folate deficiency [25] as well as MTHFR C677T polymorphism [26] are suspected as early risk factors for cognitive impairment in the elderly. Such tests could be applied as part of a cognitive test battery to assess patients with memory impairment. Utilizing the available data from a neurology-based outpatient memory clinic, we aimed to investigate the association between sUA and sHcy, as well as the potential effect of MTHFR C677T polymorphism on the sUA and sHcy levels, and on vascular disease outcomes in cognitive impaired elderly patients.

## 2. Materials and Methods

### 2.1. Study Population and Data Collection

Study participants consisted of 861 patients (597 females and 264 males) attending a neurology outpatient clinic to undergo memory impairment screening at a tertiary university hospital in Seoul, Korea, between June 2006 to May 2015. All participants underwent extensive clinical interview, neurological examination, a Korean version of the Mini-Mental State Examination (K-MMSE) [27], and a Clinical Dementia Rating (CDR) [28] to assess global functional and cognitive impairment. The patients with Alzheimer’s dementia (AD) were diagnosed according to the criteria of the National Institute of Neurological and Communicative Disorders and Stroke and the Alzheimer’s Disease and Related Disorders Association (NINCDS/ADRDA) [29]. AD with cerebrovascular disease (AD with CVD) and vascular dementia (VD) were diagnosed under the NINDS/the Association Internationale pour la Recherche et l’Enseignement en Neurosciences (AIREN) criteria for probable and possible VD [30]. Mild cognitive impairment (MCI) was diagnosed by Petersen criteria [31]. Clinical probable dementia with Lewy bodies (DLB) was diagnosed by the revised criteria for the clinical diagnosis of probable and possible DLB [32].

All of the patients were examined after overnight fasting. We excluded participants taking medication that may affect sUA level. The data allowed us to explore the correlative relationships between different variables such as sUA, sHcy, folic acid, vitamin B12 and MTHFR genotype. Clinical and laboratory data collection as well as genotyping for MTHFR C677T polymorphism were performed as a baseline study. Hyperuricemia was defined as >5.6 mg/dL for female subjects and >7.0 mg/dL for male subjects [33]. Hyperhomocysteinemia was defined as >15 μmol/L [34].

### 2.2. Statistical Analysis

The data for female and male participants were analyzed using descriptive statistics and the Mann–Whitney test. The means for all data were compared using the Mann–Whitney test; data are presented as mean ± standard deviation (SD). The correlation between the sUA and other variables was assessed using odds ratios (OR) and Pearson’s correlation analysis. In the determination of MTHFR genotypes, the Hardy–Weinberg equilibrium was examined using a chi-squared analysis. A logistic regression model was applied with adjusting for age, systolic blood pressure (SBP), body mass index (BMI), serum creatinine, triglyceride, low-density lipoprotein (LDL) cholesterol, high-density lipoprotein (HDL) cholesterol, and HbA1c. Analysis of variance (ANOVA) was used to assess the effect of the MTHFR genotype on baseline clinical and biochemical variables. The Jonckheere–Terpstra test was used for differences of the MTHFR genotype on baseline, clinical and biochemical variables among dementia sub-group. The *p* value required for statistical significance was <0.05. All statistical analyses were conducted using SPSS version 22.0 (SPSS Inc., Chicago, IL, USA).

### 2.3. Ethics Statement

The study protocol was reviewed and approved by the Institutional Review Board (IRB) of Hanyang University Hospital (IRB 2015-10-014).

## 3. Results

### 3.1. Basic Characteristics of the Study Population

A total of 861 patients (597 female and 264 male) were analyzed (Table 1). There were differences in the mean age between female and male patients (female, 75.3 ± 10.3 years vs. male, 73.2 ± 11.3 years; *p* = 0.006). The higher prevalence of cerebrovascular accidents was observed in male compared to female subjects (11.4% vs. 5.7%, *p* = 0.003). In serologic analyses, level of total cholesterol, HDL cholesterol, vitaminB12, and folic acid were low in male compare to female. Male subjects showed a significantly greater increased mean sUA level (5.7 ± 1.4 mg/dl vs. 4.6 ± 1.3 mg/dl, *p* < 0.001), as well as sHcy level (16.0 ± 11.9 μmol/L vs. 12.4 ± 7.8 μmol/L, *p* < 0.001) than female subjects.

### 3.2. Correlation between Serological Factors and Uric Acid Levels and Gender Effect

Male subjects showed a significantly increased mean sUA level (5.7 ± 1.4 mg/dL vs. 4.6 ± 1.3 mg/dL, *p* < 0.001) and sHcy level (16.0 ± 11.9 μmol/L vs. 12.4 ± 7.8 μmol/L, *p* < 0.001) than female subjects. Female subjects showed significantly lower smoking, alcohol intake, and LDL cholesterol levels, whereas HDL cholesterol, vitamin B12, and folic acid were significantly higher compared to male subjects (Table 2). A lower prevalence of cerebrovascular accidents was observed in female compared to male subjects (5.7% vs. 11.4%, *p* = 0.003).

### 3.3. Correlation between Homocystein and Uric Acid Levels and Gender Effect

Compared with subjects with normal sHcy levels, those with hyperhomocysteinemia had lower serum folic acid (7.3 ± 3.6 nmol/L vs. 11.0 ± 4.6 nmol/L) and vitamin B12 (559.2 ± 352.8 pmol/L vs. 742.0 ± 357.2 pmol/L) while having higher sUA levels (5.5 ± 1.6 mg/dL vs. 4.8.0 ± 1.3 mg/dL; *p* < 0.001 for all comparisons) (Table 3).

Pearson’s coefficient analysis showed that sUA was significantly associated with serum creatinine, HbA1c, and sHcy for female and male subjects (Table 4). Table 4 shows the results of logistic regression analysis for the relationship between elevated sHcy and hyperuricemia. Subjects with an elevated sHcy level had an odds ratio (OR) of 2.0 (95% confidence interval (CI) 1.4–3.1, *p* = 0.001) in female subjects and 2.7 (95% CI 1.4–5.4, *p* = 0.003) in male subjects for hyperuricemia (Model 1, Table 4). The association was marginally significant (female: OR, 1.0; *p* = 0.022; male: OR, 1.0; *p* = 0.014) after adjusting for age (Model 2, Table 4). After adjusting for other determinants (age, SBP, BMI, creatinine, triglyceride, LDL cholesterol, HDL cholesterol, and HbA1c), it remained significant in female subjects (OR, 1.1; 95% CI, 1.0–1.1; *p* = 0.02) but not in male subjects (OR, 1.0; 95% CI, 0.99–1.07; *p* = 0.187; Model 3, Table 4).

### 3.4. Effect of MTHFR C677T Polymorphism

The genotype frequency of MTHFR C677T was 32.8% (*n* = 282) for CC, 49.8% (*n* = 429) for CT, and 17.4% (*n* = 150) for the TT genotype. The distribution in the Hardy–Weinberg equilibrium was *p* = 0.548. In both sexes, the genotype frequencies were similar. The TT genotype showed significant association with hyperhomocysteinemia when both genders were analyzed (ANOVA, *p* = 0.001) (Table 5). However, there was no significant association between MTHFR genotypes and sUA level for female and male subjects (*p* = 0.428, *p* = 0.116, respectively). The OR of hyperuricemia to MTHFR genotypes was not significant. In female subjects OR = 1.2 (95% confidence interval, 0.8–1.8) for the CC genotype, OR = 0.9 (0.6–1.3) for the CT genotype, and OR = 1.0 (0.6–1.7) for the TT genotype; in male subjects OR = 1.6 (95% confidence interval, 0.8–3.4) for CC genotype, OR = 0.5 (0.3–1.1) for the CT genotype, and OR = 1.3 (0.5–3.2) for the TT genotype. According to analysis by dementia subgroup, there might be no significant effect of the MTHFR genotypes on the sUA level in each group.

The genotype distribution did not significantly deviate from the Hardy–Weinberg equilibrium.

### 3.5. Biochemical Characteristics of Patients According to Dementia Sub-Classification

Demographics according to dementia subgroups under clinical criteria are summarized in Table 6. There were significant differences in age, sex, MMSE and CDR among groups. In clinical parameters, systolic BP was significant high in AD with CVD and VD groups. The level of creatinine, folic acid, and sHcy were differed among dementia subtype. In particular, AD with CVD patients and VD patients showed increased mean sUA levels and hyperhomocysteinemia (Table 6). Creatinine level was increased in patients with AD with CVD, VD and DLB. An increase of folic acid level was also observed in MCI patients. MTHFR genotype was not different among dementia subgroups.

## 4. Discussion

This is the first study that examined the association between sUA level and MTHFR C677T polymorphism according to gender in patients with cognitive impairment in Korea. We postulated that MTHFR mutation results in elevated sHcy levels and high sUA levels, and they were positively associated with metabolic risk factors. Firstly, sUA is positively associated with sHcy in males. Alcohol consumption, smoking, cholesterol level, and renal function were significantly different between female and male subjects. Female and male data were analyzed separately due to these differences, which could be potential confounding factors. Male subjects showed a higher prevalence of cerebrovascular accidents as well as a higher sHcy level compared to female subjects. These results suggest that when hyperuricemia and hyperhomocysteinemia coexist could increase the vascular burden in those with cognitive impairment. Normally, adult male sUA levels generally exceed those in women of reproductive age due to the uricosuric effect of estrogen [35].

Secondly, the variant MTHFR TT genotype is associated with hyperhomocysteinemia as previously reported [36], whereas, the MTHFR C677T polymorphism is not associated with sUA levels in this study. sUA level is determined by a complex interplay of genetic and environmental factors. The association between sUA and sHcy was first reported by Kang et al. [37], and Cohen et al. [17] showed a possible link between hyperuricemia and hyperhomocysteinemia in a large retrospective study. In this study, we speculated the clinical implications of the MTHFR polymorphism and association with other biochemical factors. The polymorphic variant in MTHFR C677T has been implicated in a mild form of MTHFR deficiency linked to hyperhomocysteinemia [18]. MTHFR deficiency disrupts homocysteine metabolism. A previous study demonstrated that a mutation of MTHFR C677T increases sHcy concentration and decreases folate [38]. Hemodialysis patients with the TT and CT genotypes of the MTHFR gene have significantly greater sHcy and lower serum folate concentrations than those with the CC genotype [39]. The TT genotype had clearly higher levels of homocysteine than CT heterozygotes or CC homozygotes [40]. Recently, there has been a consensus that the homozygous TT genotype is an independent risk factor for hyperhomocysteinemia. Our results were consistent with studies finding that the TT genotype showed significant association with hyperhomocysteinemia. MTHFR catalyzes the reduction of 5,10-methylenetetrahydrofolate to 5-methyltetrahydrofolate. A c.677C > T (A222V) variant of MTHFR gene is responsible for MTHFR with reduced enzymatic activity [41]. The T allele of the MTHFR c.677C > T gene may be associated folate and vitamin B12 deficiency and hyperhomocysteinemia causing impaired one-carbon transfer (methylation) reactions, which are necessary for the production of monoamine neurotransmitters, phospholipids, and nucleotides [42]. Homocysteine may also have a direct neurotoxic effect, leading to cell death [43]. Hcy-induced vascular injuries lead to intimal thickening, elastic lamina disruption, smooth muscle hypertrophy, marked platelet dysfunction, and formation of platelet-rich occlusive thrombi, which are all associated with cardiovascular disorder [44]. Increased sUA levels have also been shown to play a key role in cardiovascular disease. Underlying mechanisms by which sUA is the final product of purine degradation with xanthine oxidase, an enzyme implicated as a mechanistic participant in oxidative stress [17], and contributes to endothelial dysfunction and increased oxidative stress within the glomerulus and the tubulointerstitium with associated increased remodeling fibrosis of the kidney and to be proatherosclerotic and proinflammatory [45]. According to these results, the average levels of sUA seem to be around <6mg/dL which did not reach the definition of hyperuricemia. The interpretation of elevated sUA should be considered cautiously depending on its circulating and biochemical environment [46]. It may be more accurate for researchers to use z-score when comparing normal and serum UA [47]. It has been known to elevated sUA associated with brain atrophy, increase white matter lesion volume, and cognition through affecting to Aß metabolism [48] and inflammatory responses to oxidative stress, endothelium dysfunction, and cerebral microvasculature damage and remodeling [46,49]. Elevated sUA would have a direct effect on the vascular supply affecting macrovessels, particularly the afferent arterioles combining with metabolic syndrome, such as hypertension, hypercholesterolemia, high glucose, and high BMI [46,49,50] in highly genetically vulnerable group with MTHFR gene mutation. According to a current review, the relationship between sUA and cognitive system remains still debate. Although the hypothesis of a neuroprotective action of UA towards cognitive system in AD and PD patients has been suggested in some reports, toxic effects of sUA would influence differently depending on dementia subtype [8].

Several studies have shown an association between elevated plasma homocysteine levels and cognitive impairment, indicating that it may play a role in the pathophysiology of dementia [51]. In addition, elevation of sHcy is a very sensitive indicator of clinical vitamin B12 deficiency and is associated with age, sex, folate deficiency, and MTHFR polymorphism [34]. In accordance with the study, our study showed significantly lower levels of folic acid and vitamin B12 in subjects with hyperhomocysteinemia compared to subjects with normal sHcy levels. There is evidence that adequate stores of folic acid and vitamin B12 are necessary for normal homocysteine metabolism for dementia patients with vascular lesion [15].

Despite high heterogeneity in the existing studies, serum uric acid may modulate cognitive function differently according to the etiology of dementia [52]. In this study, hyperhomocysteinemia and hyperuricemia were observed in dementia subtypes with vascular lesions (AD with CVD and vascular dementia). According to these result, dementia with vascular lesion (esp, AD with CVD, VD, and NPH) have a tendency to show high systolic BP, creatinine level, hyperuricemia and hyperhomocysteinemia, which have been known as vascular risk factors. The level of sHcy was significant correlated with MTHFR TT genotype. Current research proves that uric acid can act as a nerve protection against Alzheimer’s disease and Parkinson’s disease dementia, and hyperuricemia represents a risk factor that indicates the rapid progress of the disease and possible signs of malnutrition [8]. It is known that high uric acid can exert beneficial functions due to its antioxidant properties, which may be particularly relevant in the context of neurodegenerative diseases [52]. However, recent studies have suggested that these two markers are associated with an increased risk of vascular disease [13,53]. With regard to pathological background, two core pathology of AD has been suggested for abnormal amyloid-beta protein deposition and increase of hyperphosphorylated tau protein subsequently [54]. VD has been caused by ischemic or hemorrhagic vascular burden [30]. DLB also has mixed pathology with amyloid, tau, vascular burden as well as synucleinopathy [55]. Longitudinal clinical–pathological studies have increasingly recognized the importance of mixed pathologies (the coexistence of one or more neurodegenerative and cerebrovascular disease pathologies) as important factors in the development of AD and other forms of dementia. It has been suggested that impact of multiple pathologies including vascular burden have important role on the threshold for clinically overt dementia [56]. Based on this result, hyperuricemia might be related with hyperhomocysteinemia which could affect the vascular insult to a brain in various types of dementia.

Genome-wide association studies have identified common genetic factors that contribute to hyperuricemia; these are usually associated with the renal urate transport system [57]. We failed to find out that there was a relationship between the polymorphic variant of MTHFR C677T and serum uric acid level. There have been some reports of the association between the MTHFR polymorphism and sUA. MTHFR C677T polymorphism was reported to be associated with sUA level [12,58,59,60]. Some studies have suggested that the MTHFR mutation could affect mechanisms such as the de novo synthesis of purines via 10-formyl tetrahydrofolate, with a consequent overproduction of sUA by the substrate of the MTHFR reaction [61]. C677T MTHFR mutation may be a risk factor for hyperuricemia. However, other studies reported that there was no association between MTHFR C677T polymorphism and sUA levels [61,62]. Based on this study result, sUA was associated with hyperhomocystenemia, especially male. Although it failed to identify the direct link between the uric acid and the MTHFR genotype, we could see that the genetic variation of MTHFR is related to hyperhomocystenemia and might affect the vascular burden depending on the type of dementia along with the high uric acid level.

This study has some limitations. First, it was difficult to establish a prospective healthy control group; the enrolled available elderly patients were a neurology outpatient clinic-based population who complained. Some age-related conditions could contribute to changes in biochemical markers, representing a possible selection bias. Second, we did not examine the detailed medication history, which could be relevant to the potential effects of hyperuricemia. This is the first report to demonstrate the different relationship between sUA and sHcy according to gender and simultaneously evaluate MTHFR polymorphism effect in the Korean population with varying cognitive etiology, despite some limitations.

## 5. Conclusions

This study revealed that sUA is positively associated with sHcy. The novelty of our study is that the markers presumably have a synergistic effect on increasing vascular burden in cognitive decline. Controlling those levels would be good strategies to improve cognition to patient with MTHFR TT type, especially male. Additional research of the effect of sUA and sHcy is needed for various neurological disorders because it would be a modifiable risk factor by administration of supplemental folic acid and vitamin B12.

## Figures and Tables

**Table 1 nutrients-12-01147-t001:** Baseline characteristics of the study population.

	Female, *n* = 597	Male, *n* = 264	*p*-Value
Age (years)	75.3 ± 10.3	73.2 ± 11.3	0.008 *
BMI (kg/m^2^)	23.4 ± 3.4	23.5 ± 3.0	0.740
SBP (mmHg)	129.1 ± 19.1	127.4 ± 18.6	0.225
DBP (mmHg)	74.6 ± 11.3	75.0 ± 11.6	0.667
Hypertension (%)	53.4	51.5	0.603
Diabetes (%)	26.5	23.9	0.420
Smoker (%)	3.7	22.3	<0.001 *
Alcohol (%)	13.6	41.7	<0.001 *
Dyslipidemia (%)	8.4	6.1	0.239
CVD (%)	5.7	11.4	0.003 *
Cholesterol, mean ± standard deviation (SD) (mg/dL)	199.5 ± 38.9	184.9 ± 38.0	<0.0001 *
Glucose, mean ± SD (mg/dL)	111.1 ± 37.6	110.3 ± 32.1	0.685
Creatinine, mean ± SD (mg/dL)	0.8 ± 0.3	1.0 ± 0.2	<0.001 *
Triglyderide, mean ± SD (mg/dL)	130.6 ± 69.3	127.3 ± 75.4	0.090
HDL, mean ± SD (mg/dL)	51.6 ± 13.2	46.8 ± 11.8	<0.001 *
LDL, mean ± SD (mg/dL)	114.5 ± 32.4	108.0 ± 32.1	0.008
HbA1c, mean (SD) (%)	6.0 ± 1.0	6.0 ± 1.2	0.112
VitaminB12, mean (SD) (pmol/L)	727.0 ± 386.2	616.4 ± 299.0	<0.0001 *
Folic acid, mean (SD) (nmol/L)	10.5 ± 4.7	9.0 ± 4.4	<0.0001 *
Homocysteine, mean (SD) (μmol/L)	12.4 ± 7.8	16.0 ± 11.9	<0.0001 *
Uric acid, mean (SD) (mg/dL)	4.6 ± 1.3	5.7 ± 1.4	<0.0001 *
MTHFR (C677T)			
CC (%)	32.7	33.0	0.933
CT (%)	49.6	50.4	0.829
TT (%)	17.8	16.7	0.698

BMI, body mass index; SBP, systolic blood pressure; DBP, diastolic blood pressure; CVD, cerebrovascular disease; HDL, high-density lipoprotein cholesterol; LDL, low-density lipoprotein cholesterol; HbA1c, hemoglobin A1c; MTHFR, methylenetetrahydrofolate reductase; * Statistically significant at *p* < 0.05.

**Table 2 nutrients-12-01147-t002:** Correlation coefficient between different variables and uric acid levels.

	Female		Male	
	r	*p*-Value	r	*p*-Value
Age (years)	0.143	<0.001 *	0.008	0.903
Cholesterol (mg/dL)	−0.003	0.943	0.037	0.558
Glucose (mg/dL)	0.043	0.302	−0.161	0.01 *
Creatinine (mg/dL)	0.375	<0.001 *	0.408	<0.001 *
Triglyceride (mg/dL)	0.219	<0.001	0.084	0.201
HDL (mg/dL)	−0.180	<0.001	−0.023	0.729
LDL (mg/dL)	0.009	0.829	0.024	0.718
HbA1c (%)	0.090	0.038 *	−0.216	0.001 *
VitaminB12 (pmol/L)	0.032	0.434	−0.078	0.205
Folic acid (nmol/L)	0.013	0.758	0.020	0.753
Homocysteine (μmol/L)	0.174	<0.001 *	0.148	0.016 *

r Pearson’s coefficient; * Statistically significant at *p* < 0.05.

**Table 3 nutrients-12-01147-t003:** Characteristics of subjects with high and normal homocysteine levels.

	Homocysteine > 15 μmol/L, *n* = 264 (30.7%)	Homocysteine ≤ 15 μmol/L, *n* = 597 (69.3%)	*p*-Value
Vitamin B12, mean ± SD (pmol/L)	559.2 ± 352.8	742.0 ± 357.2	<0.001 *
Folic acid, mean ± SD (nmol/L)	7.3 ± 3.6	11.0 ± 4.6	<0.001 *
Uric acid, mean ± SD (mg/dL)	5.5 ± 1.6	4.8 ± 1.3	<0.001 *
MTHFR (C677T)			
CC (%)	28.7	34.2	0.126
CT (%)	50.9	49.4	0.712
TT (%)	20.4	16.3	0.159

* Statistically significant at *p* < 0.05; Hyperhomocysteinemia was defined as >15 μmol/L.

**Table 4 nutrients-12-01147-t004:** Association between elevated homocysteine serum levels and hyperuricemia.

	Female		Male	
	OR (95% CI)	*p*-Value	OR (95% CI)	*p*-Value
Model 1	2.0 (1.4–3.1)	0.001	2.7 (1.4–5.4)	0.003 *
Model 2	1.0 (1.0–1.1)	0.022	1.0 (0.97–1.0)	0.014 *
Model 3	1.1 (1.0–1.1)	0.020	1.028 (0.99–1.07)	0.187

Model 1, crude; model 2, adjustment for age; model 3, adjustment for age, SBP, BMI, Creatinine, Triglyceride, LDL cholesterol, HDL cholesterol, HbA1c. Hyperuricemia was defined as >5.6 mg/dL for female subject and >7.0 mg/dL for male subject. Hyperhomocysteinemia was defined as >15 μmol/L. * Statistically significant at *p* < 0.05;

**Table 5 nutrients-12-01147-t005:** Biometric and biochemical characteristics of patients according to the MTHFR genotypes.

	Female				Male			
	CC (*n* = 195)	CT (*n* = 296)	TT (*n* = 106)	*p*-Value	CC (*n* = 87)	CT (*n* = 133)	TT (*n* = 44)	*p*-Value
Age (years)	74.6 ± 10.6	76 ± 9.7	74.9 ± 11	0.279	72.4 ± 11.3	73.1 ± 11.7	74.8 ± 9.9	0.279
BMI (kg/m^2^)	23.2 ± 3.7	23.4 ± 3.4	23.8 ± 3.1	0.348	23.8 ± 2.9	23.4 ± 3.1	23.1 ± 2.8	0.348
Cholesterol (mg/dL)	202.3 ± 41.7	199.1 ± 36.2	195.2 ± 40.5	0.328	182.1 ± 39.1	187.6 ± 37.1	182.7 ± 38.4	0.328
Glucose (mg/dL)	109.7 ± 29	111.1 ± 40.9	114.1 ± 42.2	0.640	115.3 ± 32.1	108.5 ± 34.6	105.5 ± 22.1	0.640
Creatinine (mg/dL)	0.82 ± 0.19	0.84 ± 0.31	0.82 ± 0.17	0.631	0.97 ± 0.16	1.01 ± 0.2	1.01 ± 0.19	0.631
Triglyderide (mg/dL)	130.2 ± 77	128. 9 ± 59.6	135.9 ± 78.7	0.685	126.7 ± 73.1	130.5 ± 81.8	118.4 ± 57.5	0.685
HDL (mg/dL)	51.3 ± 12.3	52.2 ± 14.1	50.3 ± 12.2	0.435	46.2 ± 10.9	47.2 ± 12.3	46.8 ± 12.5	0.435
LDL (mg/dL)	116.4 ± 32.4	114.1 ± 31.2	112.3 ± 35.7	0.593	105.2 ± 32.6	110.7 ± 32.1	105.7 ± 31	0.593
HbA1c (%)	6.0 ± 1	6 ± 1.1	6.1 ± 1.1	0.894	6.1 ± 1.2	5.9 ± 1.2	5.9 ± 0.9	0.894
VitaminB12 (pmol/L),	727.4 ± 377	727.9 ± 420	723.6 ± 297	0.995	609.1 ± 276.3	616.4 ± 288.4	631 ± 371.9	0.995
Folic acid (nmol/L)	11 ± 4.7	10.2 ± 4.7	10.1 ± 4.6	0.130	9.6 ± 4.2	9.0 ± 4.6	7.4 ± 3.8	0.130
Homocysteine (μmol/L)	11.5 ± 3.9	12.2 ± 4.8	14.9 ± 15.6	0.001 *	14.1 ± 5.2	14.4 ± 6.5	24.2 ± 24.3	0.001 *
Uric acid (mg/dL)	4.7 ± 1.4	4.6 ± 1.3	4.7 ± 1.2	0.428	5.6 ± 1.5	5.8 ± 1.3	5.7 ± 1.3	0.428

^*^ The values in different MTHFR genotypes (means ± SD) were compared by ANOVA.

**Table 6 nutrients-12-01147-t006:** Biometric and biochemical characteristics of patients according to dementia sub-classification.

	Dementia Classification							
	AD (*n* = 374)	AD with CVD (*n* = 65)	VD (*n* = 94)	MCI (*n* = 185)	DLB (*n* = 48)	NPH (*n* = 36)	Others (*n* = 69)	*p*-Value
Age (years)	77.8 ± 9.0	81.2 ± 7.5	74.3 ± 10.6	66.7 ± 10.6	80.1 ± 7.3	75.8 ± 9.7	71.7 ± 7.2	0.000 *
Sex (F %)	74	72	51	70	77	61	55	0.016 *
BMI (kg/m^2^)	23.2 ± 3.7	23.4 ± 3.4	23.8 ± 3.1	23.8 ± 3.1	23.8 ± 2.9	23.4 ± 3.1	23.1 ± 2.8	0.348
SBP (mmHg)	129.7 ± 20.2	134.6 ± 16.2	135.3 ± 20.1	124.8 ± 15.5	129.3 ± 16.7	122.9 ± 21.1	122.6 ± 19.8	0.027 *
DBP (mmHg)	74.6 ± 11.6	76.5 ± 13.0	79.1 ± 10.1	74.5 ± 10.1	73.7 ± 11.2	71.4 ± 11.1	70.4 ± 12.1	0.224
Hypertension (%)	25.4	76.9	67.0	38.9	64.5	44.4	39.1	0.099
Diabetes (%)	24.5	30.7	41.4	16.2	27.0	30.5	21.7	0.666
Smoker (%)	8.37	6.15	13.8	10.2	4.16	19.4	7.24	0.285
Alcohol (%)	19.1	10.7	29.7	29.7	8.33	11.1	28.9	0.103
Dyslipidemia (%)	8.37	3.07	6.52	11.3	2.08	5.55	7.24	0.730
CVD (%)	4.05	13.8	22.3	3.24	4.16	11.1	10.1	0.091
Cholesterol (mg/dL)	197.5 ± 39.1	179.7 ± 34.1	189 ± 41.8	195.5 ± 37.3	199.3 ± 43.5	194.6 ± 32.8	195 ± 30.4	0.265
Glucose (mg/dL)	111.8 ± 39.1	107.5 ± 26.1	113.4 ± 36	103.8 ± 20	114.1 ± 38.2	105.6 ± 32.1	114.4 ± 22.1	0.175
Creatinine (mg/dL)	0.87 ± 0.20	0.96 ± 0.33	0.95 ± 0.51	0.81 ± 0.14	0.92 ± 0.23	0.86 ± 0.21	0.78 ± 0.19	0.004 *
Triglyderide (mg/dL)	128.9 ± 69.4	129.6 ± 68.0	139.7 ± 78.4	119.8 ± 65.8	144.3 ± 89.8	140.8 ± 76.3	138.1 ± 57.5	0.319
HDL (mg/dL)	50.2 ± 12.3	47.3 ± 9.95	47.9 ± 12.2	51.5 ± 13.6	49.0 ± 16.1	49.3 ± 13.3	50.08 ± 12.5	0.435
LDL (mg/dL)	113.9 ± 31.8	103.4 ± 29.9	108.8 ± 33.0	112.7 ± 31.8	115.3 ± 37.1	112.0 ± 26.9	112.4 ± 31.0	0.643
HbA1c (%)	6.0 ± 1.1	6.1± 1.0	6.0 ± 1.0	5.7 ± 0.7	6.0 ± 0.8	6.2 ± 1.2	5.9 ± 0.9	0.322
VitaminB12 (pmol/L)	693.7 ± 369	668.2 ± 423	701.2 ± 362	719.4 ± 315	717.7 ± 472	675.2 ± 369	687.7 ± 371	0.930
Folic acid (nmol/L)	9.87 ± 4.7	8.60 ± 4.5	9.42 ± 4.6	11.2 ± 4.5	8.61 ± 4.8	10.1 ± 5.3	9.39 ± 4.0	0.006 *
Homocysteine (μmol/L)	13.6 ± 8.4	19.7 ± 18.2	13.8 ± 6.9	10.5 ± 5.7	15.3 ± 5.7	14.4 ± 8.9	13.5 ± 3.92	0.000 *
Uric acid (mg/dL)	4.93 ± 1.4	5.20 ± 1.6	5.24 ± 1.2	4.79 ± 1.2	4.79 ± 1.4	4.87 ± 1.5	4.93 ± 1.5	0.206
MMSE	19.0 ± 6.3	17.1 ± 5.8	19.7 ± 6.6	25.3 ± 4.0	18.1 ± 5.7	18.7 ± 7.3	17.7 ± 6.3	0.000 *
CDR	1.05 ± 0.58	1.36 ± 0.63	1.13 ± 0.62	0.62 ± 0.36	1.06 ± 0.54	1.24 ± 0.70	1.26 ± 0.15	0.000 *
MTHFR								0.222
CC (%)	123 (32.9)	14 (21.7)	31 (33.0)	72 (38.9)	14 (29.1)	9 (25.0)	19 (27.5)	
CT (%)	197 (52.6)	34 (52.1)	47 (50.0)	74 (40.0)	27 (56.3)	17 (47.2)	33 (47.8)	
TT (%)	53 (14.3)	17 (26.0)	16 (17.0)	39 (21.0)	7 (14.6)	10 (27.8)	9 (13.0)	

Jonckheere–Terpstra test was used for differences of the MTHFR genotype on baseline, clinical and biochemical variables among dementia sub-group; The genotype distribution did not significantly deviate from the Hardy–Weinberg equilibrium; Abbreviations: AD, Alzheimer’s dementia; AD with CVD, Cerebrovascular disease with Alzheimer’s dementia; VD, Vascular dementia, MCI, Mild cognitive impairment; DLB, Dementia with Lewy Bodies; NPH, Normal pressure hydrocephalus; Others mean frontotemporal dementia syndrome, multiple system atrophy, and unspecified dementia. * Statistically significant at *p* < 0.05.

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
