# Peer review of "The Different Relationship between Homocysteine and Uric Acid Levels with Respect to the MTHFR C677T Polymorphism According to Gender in Patients with Cognitive Impairment"

_nutrients, 2020, doi:10.3390/nu12041147_

Round 1
Reviewer 1 Report
Line 107 – Table 2 should be replaced by table 1
Line 119 – Table 2 should be replaced by table 1
Line 129 – Table 3 should be replaced by table 2
Line 130 – Table 4 in parenthesis should be replaced by table 3
Line 174 – provide reference(s)
Though interesting, this study is too preliminary. The authors show that there is a positive correlation between sUA and sHcy levels, however, they do not underline or even postulate the possible mechanisms for this correlation. As clichéd as it sounds, but, correlation does not always mean causation and any number of reasons for this correlation can be speculated. The authors should provide more convincing data to corroborate their claim.
Author Response
Dear Editor in Chief
I would like to express our sincere gratitude for your thorough consideration and scrutiny over our manuscript. Through the accurate and keen comments made by the reviewer, the critical points at issue in the analyses of the data and the overall manuscript were discovered and subsequently corrected. After receiving the reviewers’ criticisms, my colleagues and I have extensively revised the manuscript in order to achieve the proper scientific and literary levels required by the reviewer and Nutrients. I hope this revised manuscript will be considered positively and be accepted by Nutrients.
Sincerely yours,
Hee-Jin Kim
==============================
Our responses to the reviewer’s comments are as follows:
Reviewer #1:
Comments and Suggestions for Authors
Line 107 – Table 2 should be replaced by table 1
Line 119 – Table 2 should be replaced by table 1
Line 129 – Table 3 should be replaced by table 2
Line 130 – Table 4 in parenthesis should be replaced by table 3
Line 174 – provide reference(s)
Though interesting, this study is too preliminary. The authors show that there is a positive correlation between sUA and sHcy levels, however, they do not underline or even postulate the possible mechanisms for this correlation. As clichéd as it sounds, but, correlation does not always mean causation and any number of reasons for this correlation can be speculated. The authors should provide more convincing data to corroborate their claim.
Response:
We thank the reviewer for bringing our attention to these points. The text has been modified in detail. We tried to highlight the clinical importance of modification of various nutrient factors by revealing the relationship between the patient's uric acid level and other biological markers in terms of nutrition. The arrangement and form of unraveling of the various results during the second view did not sufficiently communicate the researchers' arguments, and the results were modified. The arrangement of the results and the form of solving the logic did not sufficiently communicate our thesis, so the results were modified once again. And we provide relationship between sUA and sHcy in pathomechanism in discussion part.
Correction:
1.Subtitles have been changed.
|
3.1. Basic characteristics of the study population 3.2. Serum uric acid and serum homocysteine according to gender and dementia classification 3.3. Correlation between serum uric acid and serum homocysteine 3.4. Effect of MTHFR C677T polymorphism |
|
3.1. Basic characteristics of the study population 3.2. Correlation between serological factors and uric acid levels and gender effect 3.3. Correlation between homocystein and uric acid levels and gender effect 3.4. Effect of MTHFR C677T polymorphism 3.5. Biochemical characteristics of patients according to dementia sub-classification |
- Table renumbering
Line 107 – Table 2 should be replaced by table 1
: Table 2 became Table 4 according to the text.
Line 119 – Table 2 should be replaced by table 1
: Table 2 became Table 4 according to the text.
Line 129 – Table 3 should be replaced by table 2
: Table 3 became Table 2 according to the text.
Line 130 – Table 4 in parenthesis should be replaced by table 3
: Table 4 remained Table 4 according to the text.
Line 174 – provide reference(s)
: I proved reference #36. (Andreassi, M., et al. (2003). "Methylenetetrahydrofolate reductase gene C677T polymorphism, homocysteine, vitamin B12, and DNA damage in coronary artery disease." Human genetics 112(2): 171-177.)
I suggested the possible mechanism and some evidences in the discussion part (line 211-231).
“MTHFR catalyzes the reduction of 5,10-methylenetetrahydrofolate to 5-methyltetrahydrofolate. A c.677C>T (A222V) variant of MTHFR gene is responsible for MTHFR with reduced enzymatic activity [41]. The T allele of MTHFR c.677C>T gene may be associated folate and vitamin B12 deficiency and hyperhomocysteinemia causing impaired one-carbon transfer (methylation) reactions, which are necessary for the production of monoamine neurotransmitters, phospholipids, and nucleotides [42]. Homocysteine may also have a direct neurotoxic effect, leading to cell death [43]. Hcy induced vascular injuries lead to the intimal thickening, elastic lamina disruption, smooth muscle hypertrophy, marked platelet dysfunction, and formation of platelet-rich occlusive thrombi, which are all associated with cardiovascular disorder [44]. Increased sUA levels have also been shown to play a key role in cardiovascular diseases underlying mechanism which sUA is the final product of purine degradation with xanthine oxidase, an enzyme implicated as a mechanistic participant in oxidative stress[45], and contributes to endothelial dysfunction and increased oxidative stress within the glomerulus and the tubulointerstitium with associated increased remodeling fibrosis of the kidney and to be proatherosclerotic and proinflammatory [46]. Elevated sUA This would have a direct effect on the vascular supply affecting macrovessels, particularly the afferent arterioles combining with metabolic syndrome, such as hypertension, hypercholesterolemia, high glucose, and high BMI [47] in highly genetically vulnerable group with MTHFR gene mutation.”

Reviewer 2 Report
This manuscript entitled “The different relationship between homocysteine and uric acid levels with respect to the MTHFR C677T polymorphism according to gender in patients with cognitive impairment” has been reviewed. The authors described the clinical significance of sUA and MTHFR C677T in elderly male and female patients with cognitive impairment. I felt that the work was of interest and well-documented.
Below I would like to list some of my concerns;
- It was still unclear whether there were statistical differences between with and without cognitive impairment in several markers/factors.
- Differences of causes of the cognitive impairment in subjects of this study, i.e. with or without Alzheimer’s dementia, should be discussed more in depth. Also, why does sUA or MTHFR C677T effect to cognitive impairment? The pathological mechanisms should be discussed.
- Did the authors confirm the inter-examiner error? Authors need to indicate if the examiners were standardized and how many examiners performed the clinical exams.
- Probably, I think the data were not normally distributed. Was Student’s t-test appropriate methods for statistical analyses?
Author Response
Dear Editor in Chief
I would like to express our sincere gratitude for your thorough consideration and scrutiny over our manuscript. Through the accurate and keen comments made by the reviewer, the critical points at issue in the analyses of the data and the overall manuscript were discovered and subsequently corrected. After receiving the reviewers’ criticisms, my colleagues and I have extensively revised the manuscript in order to achieve the proper scientific and literary levels required by the reviewer and Nutrients. I hope this revised manuscript will be considered positively and be accepted by Nutrients.
Sincerely yours,
Hee-Jin Kim
==============================
Our responses to the reviewer’s comments are as follows:
Response:
We thank the reviewer for bringing our attention to these points. The text has been modified in detail. We tried to highlight the clinical importance of modification of various nutrient factors by revealing the relationship between the patient's uric acid level and other biological markers in terms of nutrition. The arrangement and form of unraveling of the various results during the second view did not sufficiently communicate the researchers' arguments, and the results were modified. The arrangement of the results and the form of solving the logic did not sufficiently communicate our thesis, so the results were modified once again. And we provide relationship between sUA and sHcy in pathomechanism in discussion part.
Correction:
- It was still unclear whether there were statistical differences between with and without cognitive impairment in several markers/factors.
Response:
First of all, thank you for your careful review.
The study was conducted on a group of patients with cognitive disabilities who visited the cognitive disability clinic. Targets with normal cognitive function were not registered for in this study.
- Differences of causes of the cognitive impairment in subjects of this study, i.e. with or without Alzheimer’s dementia, should be discussed more in depth. Also, why does sUA or MTHFR C677T effect to cognitive impairment? The pathological mechanisms should be discussed.
Response: We provided detailed discussion focusing on the MTHFR C677T allele effect and mechanisms in various dementia. In discussion part.
(line 211-231) (line 238-263)
“MTHFR catalyzes the reduction of 5,10-methylenetetrahydrofolate to 5-methyltetrahydrofolate. A c.677C>T (A222V) variant of MTHFR gene is responsible for MTHFR with reduced enzymatic activity [41]. The T allele of MTHFR c.677C>T gene may be associated folate and vitamin B12 deficiency and hyperhomocysteinemia causing impaired one-carbon transfer (methylation) reactions, which are necessary for the production of monoamine neurotransmitters, phospholipids, and nucleotides [42]. Homocysteine may also have a direct neurotoxic effect, leading to cell death [43]. Hcy induced vascular injuries lead to the intimal thickening, elastic lamina disruption, smooth muscle hypertrophy, marked platelet dysfunction, and formation of platelet-rich occlusive thrombi, which are all associated with cardiovascular disorder [44]. Increased sUA levels have also been shown to play a key role in cardiovascular diseases underlying mechanism which sUA is the final product of purine degradation with xanthine oxidase, an enzyme implicated as a mechanistic participant in oxidative stress[45], and contributes to endothelial dysfunction and increased oxidative stress within the glomerulus and the tubulointerstitium with associated increased remodeling fibrosis of the kidney and to be proatherosclerotic and proinflammatory [46]. Elevated sUA This would have a direct effect on the vascular supply affecting macrovessels, particularly the afferent arterioles combining with metabolic syndrome, such as hypertension, hypercholesterolemia, high glucose, and high BMI [47] in highly genetically vulnerable group with MTHFR gene mutation.”
“Despite high heterogeneity in the existing studies, serum uric acid may modulate cognitive function differently according to the etiology of dementia [49]. Current research proves that uric acid can act as a nerve protection against Alzheimer's disease and Parkinson's disease dementia, and hyperuricemia represents a risk factor that indicates the rapid progress of the disease and possible signs of malnutrition [8]. In this study, hyperhomocysteinemia and hyperuricemia were observed in dementia subtypes with vascular lesions (AD with CVD and vascular dementia). It has known that high uric acid can exert beneficial functions due to its antioxidant properties, which may be particularly relevant in the context of neurodegenerative diseases [49]. However, recent studies have suggested that these two markers are associated with an increased risk of vascular disease [50,51]. Based on this result, hyperuricemia might be related with hyperhomocysteinemia which could affect vascular insult to a brain. In the view of pathological background, two core pathology of Alzheimer's disease (AD) has been suggested abnormal amyloid-beta protein deposition and increase of hyperphosphorylated tau protein subsequently [52]. VD has been caused by ischemic or hemorrhagic vascular burden [30]. DLB also has mixed pathology with amyloid, tau, vascular burden as well as synucleinopathy [53]. It has been suggested that impact of multiple pathologies including vascular burden have important role on the threshold for clinically overt dementia [54].”
- Did the authors confirm the inter-examiner error? Authors need to indicate if the examiners were standardized and how many examiners performed the clinical exams.
Response:
The study was conducted on patients who visited memory clinic which was controlled by one experienced neurologist by training in a tertiary university hospital between June 2006 to May 2015. All participants were examined one doctor.
- Probably, I think the data were not normally distributed. Was Student’s t-test appropriate methods for statistical analyses?
Response:
Following the reviewer's comments, we explored the data to see if it was distributed normally again, and all data were not normally distributed. So we performed statistical analyses from Student’s t-test to Mann-Whitney test. Jonckheere-Terpstra test was used for differences of the MTHFR genotype on baseline, clinical and biochemical variables among dementia sub-group.

Reviewer 3 Report
Dear Authors,
many thanks for Your article.
The topic is certainly intriguing, and I read your article with great attention.
I found a good article, even if limitations were relevant (as You Yourselves highlighted).
Minor suggestions :
1) some references (no. 3, 23, 25, for example) should be supplemented by more recent references;
2) what You wrote in lines 249-255 needs to be clarified, because they seemed nonsense sentences.
Author Response
I would like to express our sincere gratitude for your thorough consideration and scrutiny over our manuscript. The references have been changed more recent papers. Lines 249-255 have been modified with more clarified sentences as below.
1) some references (no. 3, 23, 25, for example) should be supplemented by more recent references;
Newland, H. Hyperuricemia in coronary, cerebral and peripheral arterial disease: an explanation. Medical hypotheses 1975, 1, 152-155.
-à
- Trifiro, G.; Morabito, P.; Cavagna, L.; Ferrajolo, C.; Pecchioli, S.; Simonetti, M.; Bianchini, E.; Medea, G.; Cricelli, C.; Caputi, A.P., et al. Epidemiology of gout and hyperuricaemia in Italy during the years 2005-2009: a nationwide population-based study. Annals of the rheumatic diseases 2013, 72, 694-700, doi:10.1136/annrheumdis-2011-201254.
# 23, 25
- Kolz, M.; Johnson, T.; Sanna, S.; Teumer, A.; Vitart, V.; Perola, M.; Mangino, M.; Albrecht, E.; Wallace, C.; Farrall, M., et al. Meta-analysis of 28,141 individuals identifies common variants within five new loci that influence uric acid concentrations. PLoS Genet 2009, 5, e1000504, doi:10.1371/journal.pgen.1000504.
- Levin, B.L.; Varga, E. MTHFR: Addressing Genetic Counseling Dilemmas Using Evidence-Based Literature. Journal of genetic counseling 2016, 25, 901-911, doi:10.1007/s10897-016-9956-7.
- Wei, W.; Liu, S.Y.; Zeng, F.F.; Ma, L.; Li, K.S.; Wang, B.Y. Meta-analysis of the association of the C677T polymorphism of the methylenetetrahydrofolate reductase gene with hyperuricemia. Annals of nutrition & metabolism 2012, 60, 44-51, doi:10.1159/000335698.
- Quadri, P.; Fragiacomo, C.; Pezzati, R.; Zanda, E.; Forloni, G.; Tettamanti, M.; Lucca, U. Homocysteine, folate, and vitamin B-12 in mild cognitive impairment, Alzheimer disease, and vascular dementia. The American journal of clinical nutrition 2004, 80, 114-122, doi:10.1093/ajcn/80.1.114.
2) what you wrote in lines 249-255 needs to be clarified, because they seemed nonsense sentences.
“In the view of pathological background, two core pathology of AD has been suggested abnormal amyloid-beta protein deposition and increase of hyperphosphorylated tau protein subsequently [53]. VD has been caused by ischemic or hemorrhagic vascular burden [31]. DLB also has mixed pathology with amyloid, tau, vascular burden as well as synucleinopathy [54]. Longitudinal clinical–pathological studies have increasingly recognized the importance of mixed pathologies (the coexistence of one or more neurodegenerative and cerebrovascular disease pathologies) as important factors in the development of AD and other forms of dementia. It has been suggested that impact of multiple pathologies including vascular burden have important role on the threshold for clinically overt dementia [55]. Based on this result, hyperuricemia might be related with hyperhomocysteinemia which could affect vascular insult to a brain in various dementia.”

Round 2
Reviewer 1 Report
The authors have addressed all my concerns and suggestions.
Author Response
Thank you for your careful review.
Reviewer 2 Report
Thank you very much for your revision. However, to investigate the conditions of patients with cognitive inpairment, authors must be used the groups of patients without cognitive impairment as suitable controls. If authors re-analyze using suitable controls, the results wolud be emphasized more and the readers would be agreed your new insight.
Author Response
The study results came from retrospectively analyzed the clinical data of patients who visited the cognitive clinic from 2006 to 2015. Even if there are no abnormalities in objective cognitive testing, the patients visiting a cognitive impairment clinic cannot be normal group because it is defined as a ‘subjective cognitive disorder’. With the same theme, we will prospectively check the results of the research in the normal elderly group.
